# Potential Stereoselective Binding of *Trans*-(±)-Kusunokinin and *Cis*-(±)-Kusunokinin Isomers to CSF1R

**DOI:** 10.3390/molecules27134194

**Published:** 2022-06-29

**Authors:** Chompunud Chompunud Na Ayudhya, Potchanapond Graidist, Varomyalin Tipmanee

**Affiliations:** Department of Biomedical Sciences and Biomedical Engineering, Faculty of Medicine, Prince of Songkla University, Songkhla 90110, Thailand; 6310330001@psu.ac.th

**Keywords:** *trans*-(−)-kusunokinin, kusunokinin isomers, CSF1R inhibitor, stereoselectivity, virtual screening, molecular dynamics simulation

## Abstract

Breast cancer cell proliferation and migration are inhibited by naturally extracted *trans*-(−)-kusunokinin. However, three additional enantiomers of kusunokinin have yet to be investigated: *trans*-(+)-kusunokinin, *cis*-(−)-isomer and *cis*-(+)-isomer. According to the results of molecular docking studies of kusunokinin isomers on 60 breast cancer-related proteins, *trans*-(−)-kusunokinin was the most preferable and active component of the *trans*-racemic mixture. *Trans*-(−)-kusunokinin targeted proteins involved in cell growth and proliferation, whereas the *cis*-(+)-isomer targeted proteins involved in metastasis. *Trans*-(−)-kusunokinin targeted CSF1R specifically, whereas *trans*-(+)-kusunokinin and both *cis*-isomers may have bound AKR1B1. Interestingly, the compound’s stereoisomeric effect may influence protein selectivity. CSF1R preferred *trans*-(−)-kusunokinin over *trans*-(+)-kusunokinin because the binding pocket required a ligand planar arrangement to form a π-π interaction with a selective Trp550. Because of its large binding pocket, EGFR exhibited no stereoselectivity. MD simulation revealed that *trans*-(−)-kusunokinin, *trans*-(+)-kusunokinin and pexidartinib bound CSF1R differently. Pexidartinib had the highest binding affinity, followed by *trans*-(−)-kusunokinin and *trans*-(+)-kusunokinin, respectively. The *trans*-(−)-kusunokinin-CSF1R complex was found to be stable, whereas *trans*-(+)-kusunokinin was not. *Trans*-(±)-kusunokinin, a potential racemic compound, could be developed as a selective CSF1R inhibitor when combined.

## 1. Introduction

Breast cancer is one of the leading causes of mortality in women worldwide. In 2020, breast cancer was estimated at 2.26 million new cases and 0.68 million deaths [1]. The common therapeutic options for breast cancer patients are surgery, radiotherapy, chemotherapy, or hormonal therapy, in combination with targeted medication [2]. Despite the more accurate methods of treatment and diagnosis being developed, the incidence and death rate of breast cancer continues to rise [1,2,3,4]. Due to the drug resistance mechanism and the high heterogeneity of breast cancers, additional targets are needed to be explored and highly specific drugs are continuously in need of investigation [2,4,5,6,7]. The development of more effective targeted therapeutic approaches could serve as a new paradigm for breast cancer treatment in the personalized medicine era.

Breast cancer cells display distinct patterns such as excessive production of certain distinctive proteins on the cell surface [6,7]. Human epidermal growth factor receptor (HER2) or ERBBR2 is one of the most common cell-surface targets for treating breast cancer [7]. Tucatinib, an ATP-competitive, small-molecule tyrosine kinase inhibitor, is a new approach by the US Food and Drug Administration (FDA) to treat advanced breast cancer in a patient who is HER2-positive in combination with trastuzumab (Herceptin) and capecitabine (Xeloda) [8,9,10,11]. Colony-stimulating factor 1 receptor (CSF1R) is another emerging cell surface target for breast cancer [12,13,14]. CSF1R is responsible for cancer cell proliferation, invasion and survival [15,16,17,18,19,20,21]. CSF1R is strongly related to poor outcomes and contributes to tumor cell invasiveness and pro-metastatic behavior [12,15,18,22,23,24,25,26]. Pexidartinib, a pyrrolopyridine-based compound, is the first FDA-approved CSF1R tyrosine kinase inhibitor [27,28]. Moreover, several pyrrolopyridine-based selective CSF1R inhibitors are developed and currently used in clinical trials [14,29,30], i.e., sotuletinib [31], ARRY-382 [32] and JNJ-40346527 [33,34,35]. Nowadays, multi-targeted therapy has also become of interest in drug development to overcome the limitation of single-target drugs [36]. For example, sorafenib is a pyridinecarboxamide protein kinase inhibitor that targets many protein kinases, including VEGFR, PDGFR and RAF kinases [37].

*Trans*-(−)-kusunokinin, natural occurring (3R,4R)-3-(1,3-benzodioxol-5-ylmethyl)-4-[(3,4-dimethoxyphenyl)methyl]oxolan-2-one from *Piper nigrum*, inhibits breast, colorectal and lung cancer cell lines and inhibits tumor growth in vivo by inhibiting topoisomerase II, PI3K, AKT, ERK1/2, cyclinB1, CDK1 and cMyc and inducing apoptosis via multi-caspase [38,39,40]. Unlike naturally extracted *trans*-(−)-kusunokinin, synthetic kusunokinin yields racemic mixtures of *trans*-(−)- and *trans*-(+)-kusunokinin or (3S,4S)-3-(1,3-benzodioxol-5-ylmethyl)-4-[(3,4-dimethoxyphenyl)methyl]oxolan-2-one, as illustrated in Figure 1A,B. *Trans*-(±)-kusunokinin inhibits topoisomerase II, STAT3, RAS, ERK, cyclinD1 and CDK1 expression in breast cancer cell lines [41,42]. The previous study hypothesized that the *trans*-(−)-isoform was the preferred active form in the *trans* racemic mixture [43], binding CSF1R at the juxtamembrane region (JM) in the same way that pexidartinib does, resulting in suppression of CSF1R and its downstream molecules on breast cancer cells [42]. Other *trans*-(−)-kusunokinin targets include mmp12, Hsp90a, cyclinB1, MEK1, MEK2 and AKR1B1, whereas the *trans*-(+)-isoform was predicted to bind HER2, CDK4 and PI3K, but its binding energies did not exceed those of the *trans*-(−)-isoform in any docked proteins [42,43,44].

*Cis*-(±)-kusunokinin isomers have been found in *Virola sebifera* [45] and *Acanthopanax chiisanensis* [46] and have also been synthesized [47]. The *cis*-(−)- and *cis*-(+)-isomers were defined from (3R,4S)- and (3S,4R)-3-(1,3-benzodioxol-5-ylmethyl)-4-[(3,4-dimethoxyphenyl)methyl]oxolan-2-ones, depicted in Figure 1C,D. *Cis*-(−)-isomer inhibits Ishikawa cells (endometrial adenocarcinoma) and has anti-estrogenic and alkaline phosphatase activities [46]. Despite the fact that the *cis*-(±)-kusunokinin isomers have a significant effect on biological activity, these isomers remain an intriguing area for drug development.

In this study, we identified the target of four kusunokinin enantiomers (*trans*-(−), *trans*-(+), *cis*-(−) and *cis*-(+)-isomer) on 60 candidate proteins related to the CSF1R-breast cancer progression pathway and proposed a preferable form of kusunokinin isomers using molecular docking and our scoring system. The potential target protein was then simulated via molecular dynamics (MD) for *trans*-(−)-kusunokinin or *trans*-(+)-kusunokinin to determine the binding mode and stereoisomeric selectivity of the target to each isomer within the *trans*-racemic mixture.

## 2. Results

### 2.1. Target Protein(s) Screening of Four Kusunokinin Isomers among CSF1R-Related Breast Cancer Progression Proteins

To investigate the possible target, binding ability and binding character of each kusunokinin stereoisomer, 60 candidate proteins related to CSF1R-associated breast cancer progression were selected for a molecular docking study. The proteins were grouped based on their primary mechanism: anti-apoptosis and survival (7 proteins), cell growth and proliferation (21 proteins), metastasis (11 proteins) and signal molecules (21 proteins). All PDB codes of the candidate proteins and PubChem CIDs for all given inhibitors were summarized in Appendix A. The docking score in kcal/mol of each kusunokinin isomer and a known inhibitor of each protein are shown in Table 1. The target selection criteria were based on a protein(s) that (1) had a lower docking score than a known inhibitor and (2) had a ligand that was located at a similar site to the known inhibitor. The docked pose and the aforementioned criteria were used to determine each isomer’s potential inhibition.

*Trans*-(−)-kusunokinin displayed a lower docking score than known inhibitors in 37 out of 60 proteins. However, *trans*-(−)-kusunokinin showed similar binding sites to the known inhibitor of 12 proteins, including anti-apoptosis and survival (COX2, Hsp90a and Hsp90b), cell growth and proliferation (cdc25A, CDK2, RagC, CSF1R and EGFR) and metastasis (ALP, PALP, AKR1B1, JAK3, MNK2 and p38) (docking score of −11.84 to −6.92 kcal/mol). The binding ability of *trans*-(−)-kusunokinin with COX2, Hsp90b, RagC, CSF1R, EGFR and p38 was the highest among the four kusunokinin isomers. *Trans*-(−)-kusunokinin had the greatest potential on CSF1R, with a docking score of −11.84 kcal/mol, whereas the CSF1R known inhibitor, pexidartinib, had a docking score of −11.59 kcal/mol.

Meanwhile, *trans*-(+)-kusunokinin showed a lower docking score than known inhibitors in 28 proteins. Seven proteins passed the potency inhibition criteria, namely COX2, Hsp90a (anti-apoptosis and survival), CDK2 and EGFR (cell growth and proliferation), ALP, PALP and AKR1B1 (metastasis), JAK3 and MNK2 (signal molecules). However, when compared to other kusunokinin stereoisomers, the *trans*-(+)-isoform did not show the best binding possibility in any of these proteins. *Trans*-(+)-kusunokinin showed the highest possibility on only AKR1B1, with a docking score of −11.15 kcal/mol, whereas epalrestat (known inhibitor) had a docking score of −10.14 kcal/mol.

The *cis*-(−)-isomer had a lower docking score than known inhibitors in 34 of 60 proteins, implying that the *trans*-(−)-kusunokinin and *cis*-(−)-isomer have remarkably similar targets for potential inhibition. Based on our criteria, the potential *cis*-(−)-isomer target proteins were COX2, Hsp90a, Hsp90b, cdc25A, CDK2, RagC, EGFR, ALP, PALP, KR1B1, JAK3, MNK2 and p38, all of which are also target proteins for *trans*-(−)-kusunokinin. Interestingly, the docking scores of the *cis*-(−)-isomer were better than those of the *trans*-(−)-isomer in CDK2, PALP, AKR1B1, JAK3 and MNK2. With a docking score of −11.46 kcal/mol, the *cis*-(−)-isomer had the highest possibility on AKR1B1.

The *cis*-(+)-isomer, on the other hand, demonstrated a lower docking score than known inhibitors in 31 proteins. COX2, Hsp90a, cdc25A, CDK2, EGFR, AKR1B1, JAK3, MNK2 and p38 were the only nine proteins that met both criteria. The *cis*-(+)-isomer also had the highest binding ability in CDK2, Hsp90a and AKR1B1 when compared to other kusunokinin isomers (docking score of −9.90, −10.77 and −11.59 kcal/mol, respectively).

Overall, 12 of 60 proteins were identified as shared potential inhibitory targets of four possible kusunokinin enantiomers, with five of them belonging to cell growth and proliferation, including cdc25A, CDK2, RagC, CSF1R and EGFR. Others were from anti-apoptosis and survival (COX2, Hsp90a and Hsp90b), metastasis (AKR1B1) and signaling proteins (JAK3, MNK2 and p38).

### 2.2. Docking Score Interpretation of the Kusunokinin Isomers

The docking scores of each kusunokinin isomer were re-analyzed in rank scores ranging from 4 to 1 using our criteria. Score 4 represented the highest binding affinity of the four isomers, followed by 3 and 2 and 1 was the lowest. The kusunokinin isoform rank scores were then multiplied by the protein’s weight score, which ranged from 3 to 1 based on drug-accessibility prediction. Number 3 stood for membrane-bound proteins, which have the highest chance of drug exposure. Number 2 represented cytoplasmic proteins, whereas number 1 represented proteins inside the nucleus, which have the lowest chance of drug exposure (Figure 2) [36,48,49,50,51]. The most preferred form was determined by adding the weighted rank scores within the protein group, which is represented as a sum score (weighted) in Table 2.

*Trans*-(−)-kusunokinin presented the highest sum score and weighted sum score in all four protein groups including anti-apoptosis and survival, cell growth and proliferation, metastasis and signaling proteins with a weighted score of 54, 151, 52 and 148, respectively. Meanwhile, *trans*-(+)-kusunokinin had the lowest sum score in all four protein groups and the lowest weighted sum score in all groups except signaling molecules.

The sum scores of *cis*-(−)- and *cis*-(+)-isomers were not the highest in any protein groups. However, the *cis*-(+)-kusunokinin isomer gave the highest weighted sum score in the metastasis proteins (weighted score of 56). No significant variation in the weighted sum scores of *cis*-(−)- and *cis*-(+)-kusunokinin isomers was observed in the anti-apoptosis and survival or the cell growth and proliferation proteins.

Based on our criteria, we proposed that *trans*-(−)-kusunokinin was the best stereoisomer for anti-apoptosis and survival, cell growth and proliferation, metastasis and signaling proteins, whereas *cis*-(+)-isomer was a potential candidate for the inhibition of metastasis proteins.

### 2.3. Target Selection Based on Ligand-Protein Interaction Analysis

The binding energies of some proteins differed between four kusunokinin isomers, as shown in Table 1, whereas others did not. One-way ANOVA was used to test the variability of the docking score among isomers within the same protein (Appendix A). Of the 12 possible targets, CSF1R had the most variety (SD = 1.089, *p* < 0.01), followed by Hsp90b and cdc25A (SD = 0.52 and 0.48, *p* < 0.01, respectively). Although the variances of EGFR, MNK2, JAK3 and AKR1B1 were all within the same range (SD = 0.032 to 0.036, *p* < 0.01), EGFR had the smallest differences between the binding energies of the four kusunokinin isomers. The 12 potential targets’ binding pockets were visualized for further investigation. Figure 3 depicts the docking poses of four kusunokinin isomers to each protein binding site.

With the exception of CSF1R and EGFR, we discovered that all 12 target pockets were relatively round and had fewer aromatic residues than CSF1R, allowing kusunokinin stereoisomers to bend to their preferred poses and fit in more freely than CSF1R, which had a narrow binding site (Figure 3). Although both CSF1R and EGFR have a “glove-like” pocket (Figure 3M), CSF1R’s narrow pocket forced kusunokinin stereoisomers into a planar arrangement in order to fit in the binding site (Figure 3A), whereas EGFR’s binding site was wider than CSF1R’s, allowing for binding arrangements without stereoselectivity (Figure 3A,L). The protein selectivity was explained by the narrow, glove-shaped CSF1R binding pocket. CSF1R and EGFR were chosen as models for studying kusunokinin stereoisomeric effects because they had the highest selectivity toward the *trans*-(−)-isoform, whereas EGFR had no selectivity to any kusunokinin stereoisomer.

CSF1R and EGFR were tyrosine kinase receptors that regulated breast cancer cell growth and proliferation [52,53,54]. Both targets revealed that the best binding possibility was to the *trans*-(−)-isomer, followed by the *cis*-(−)-, *cis*-(+)- and *trans*-(+)-isomers. Figure 4 and Figure 5 show the CSF1R and EGFR binding positions of four kusunokinin isoforms and a known inhibitor, respectively. Appendix A depicts the alignment of all docked ligand poses at the binding pocket of CSF1R or EGFR. The docked poses of pexidartinib-CSF1R and erlotinib-EGFR were found to be in good agreement with the experimental crystal structure, as shown in Appendix A. In addition, the interaction of all kusunokinin isomers as well as the inhibitor with CSF1R and EGFR was summarized in Table 3.

CSF1R provided π-π stacking with *trans*-(−)-kusunokinin, *cis*-(−)-isomer and pexidartinib via Trp550 (Figure 4A,C,E) and hydrogen bonds with Arg549 and at the same time forced *trans*-(+)-kusunokinin and *cis*-(+)-isomer to make contacts with outer residues (Figure 4B,D). On the contrary, hydrogen bonds were the most common type of interaction found between EGFR residues and ligands. The four isoforms and a known inhibitor, erlotinib, bind to EGFR in the same pocket and share Phe856 as a mutual residue (Figure 5). The EGFR binding site was larger and contained fewer aromatic residues than the CSF1R binding site, allowing all four kusunokinin structures to interact with no stereo restriction.

Two *trans*-kusunokinin isomers were aligned to compare binding position and ligand-receptor interaction, as with CSF1R and EGFR (s). The results clearly demonstrated CSF1R selectivity for *trans*-(−)-kusunokinin (Figure 6A). We also found that *trans*-(−)-kusunokinin and pexidartinib bind in similar ways, including ligand position and interaction modes (Figure 6B).

A comparison of two *trans*-kusunokinin isomers in EGFR binding was also described. Both ligands interacted with the same residues and had a strikingly similar binding position (Figure 7A). Erlotinib was also found to be related to *trans*-(−)-kusunokinin. Both ligand binding pockets were discovered to be identical but in different positions. However, when it came to forming hydrogen bonds with the residues Lys754, Asp855 and Phe856, the two ligands interacted similarly (Figure 7B). Interestingly, although erlotinib was found to form π-sulfur with Met766 and Met790 residues, the binding affinity was not as high as that of kusunokinin isomers.

### 2.4. Molecular Dynamic Simulations and Relative Binding Affinities of Trans-(±)-Kusunokinin and Pexidartinib to CSF1R

According to previous findings, CSF1R clearly prefers *trans*-(−)-kusunokinin over *trans*-(+)-kusunokinin. Therefore, the two trans isoforms were chosen for MD simulations on CSF1R in order to assess the stereoisomeric effect on protein selectivity. The structures of *trans*-(−)-kusunokinin and *trans*-(+)-kusunokinin are available in the PubChem database and have been tested for anticancer activity against breast cancer cell lines, whereas the *cis*-(±)-isomers are not yet available. MD simulations were run on *trans*-(−)-kusunokinin-CSF1R, *trans*-(+)-kusunokinin-CSF1R, pexidartinib-CSF1R and ligand-free CSF1R models in isotonic 0.15 M NaCl solution at pH 7 and 25 °C in our study. The residue number was rearranged using the AMBER20 procedure and the reference residue number is shown in Appendix A.

The root-mean square displacement (RMSD) of the 300 ns-CSF1R MD simulations was calculated to assess a structure’s conformational stability during the simulation. All MD simulations were stable, according to the RMSD plot (Figure 8A). The structure flexibility of ligand-bound and ligand-free CSF1Rs was also interpreted using the root-mean square fluctuation (RMSF) (Figure 8B). Each amino acid α carbon atom in the CSF1R structure was used to create the RMSF. To detect and compare the effect of each bound ligand on protein flexibility, the patterns of four RMSF models were plotted together. The flexibility patterns of *trans*-(−)-kusunokinin differed from those of ligand-free CSF1R from residues 10th to 50th and 130th to 150th. The overall flexibility patterns of *trans*-(−)-kusunokinin, *trans*-(+)-kusunokinin and pexidartinib were distinct, indicating that these three ligands have different effects on CSF1R conformational changes.

To investigate protein-ligand dissociation, the distance between the protein’s center of mass and the ligand at each time point was presented in Angstrom (Å) and plotted against simulation time (Figure 9A). The three ligand-CSF1R models were also visualized at 0, 100, 200 and 300 ns (Figure 9B). The distance analysis of the MD trajectories revealed that as simulation time was increased, *trans*-(+)-kusunokinin moved away from the binding pocket, whereas *trans*-(−)-kusunokinin and pexidartinib stably bound CSF1R. These findings indicated that *trans*-(−)-kusunokinin was the only isoform in the racemic mixtures that bound to CSF1R, not *trans*-(+)-kusunokinin.

The relative binding free energy was calculated using the MM/GBSA and MM/PBSA methods based on the MD trajectory (Table 4). The relative binding ability of synthetic *trans*-(±)-kusunokinin to CSF1R was measured in comparison to the relative binding free energy of pexidartinib. Pexidartinib had the highest binding ability of the selected ligands, followed by *trans*-(−)-kusunokinin and *trans*-(+)-kusunokinin. When comparing the two isomers within the *trans*-racemic mixtures, the results showed that CSF1R preferred *trans*-(−)-kusunokinin to *trans*-(+)-kusunokinin.

### 2.5. Conformational Effects of Trans-(±)-Kusunokinin and Pexidartinib on CSF1R

The distance measured from the α carbon of each CSF1R residue to the protein center of mass was expressed as a *d* value in Angstrom (Å). Then, using the following equation, the difference in distance (∆*d*) between ligand-bound and ligand-free CSF1R residues was calculated:∆*d* = *d*_*(n, ligand-bound)*_ − *d*_*(n, ligand-free)*_,(1)
when *d_(n, ligand_*_-*bound)*_ was a distance between the nth amino acid in ligand-bound CSF1R to the center of mass, and *d_(n, ligand_*_-*free)*_ was a distance between the nth amino acid in ligand-free CSF1R to the center of mass. An interpretation of the ∆*d* value is as follows:The ∆*d* value > 0 implied the residue moved further away from the protein center.The ∆*d* value < 0 implied the residue moved closer to the protein center.The ∆*d* value = 0 implied the residue stayed in the same position.

The ∆*d* values that diverged more than 2 Å were considered significant.

Figure 10 depicts the ∆*d* patterns of the three ligand-CSF1R models. *Trans*-(−)-kusunokinin was found to have a contraction effect on the residues 1st and 5th of CSF1R juxtamembrane region (JM), and it simultaneously moved the residues 135th to 145th in the kinase domain away from the reference position. The remaining ∆*d* patterns were deemed insignificant (Figure 10A).

*Trans*-(+)-kusunokinin had a similar effect on residues 135th to 145th. However, it did not have the same effect on JM conformation as *trans*-(+)-kusunokinin. Unlike *trans*-(−)-kusunokinin, the *trans*-(+)-isoform affected the activation loop (AL) residues 226th and 229th (Figure 10B).

Pexidartinib influenced the position of CSF1R residues 59th and 274th, as well as residues 135th to 145th. It also displayed similar distance patterns at the JM and AL as *trans*-(−)-kusunokinin. Nonetheless, the majority of the ∆*d* patterns were distinct, implying that both compounds affected CSF1R conformational changes differently, leading to the conclusion that *trans*-(−)-kusunokinin affected the CSF1R conformation differently than pexidartinib (Figure 10C).

## 3. Discussion

*Trans*-(−)-kusunokinin, an active compound from *P**. nigrum*, inhibited cell growth and proliferation. It was cytotoxic on luminal A breast cancer cells, triple-negative breast, colon and lung cancer cells [39,40]. *Trans*-(−)-kusunokinin was also studied for its anticancer properties in NMU-induced rat mammary tumors [40]. This natural compound bound and inhibited CSF1R [41,42]; nonetheless, siRNA-mediated CSF1R protein suppression indicated that the action of *trans*-(±)-kusunokinin differed from pexidartinib [42]. As a result, we hypothesized that *trans*-(±)-kusunokinin could bind a molecule other than CSF1R, possibly AKR1B1 [42,44].

Furthermore, *trans*-(−)-kusunokinin binding energies were lower than *trans*-(+)-kusunokinin binding energies in every protein studied, indicating that the *trans*-(−)-isomer was a potent form in the racemic compounds [43]. The different actions of extracted *trans*-(−)-kusunokinin and synthetic *trans*-(±)-kusunokinin were observed and hypothesized to be due to racemic compound nature and purity [41,42]. Because most biological target sites require stereospecific recognition of the binding counterpart, stereochemistry is one of the factors that affect pharmacokinetics and pharmacodynamics [55,56,57]. Different chiral carbon configurations in kusunokinin may result in a stereoisomeric preference for target protein selection and binding mode from all four kusunokinin isomers.

In this study, we first used molecular docking to investigate target proteins of four kusunokinin stereoisomers, followed by molecular dynamics (MD). COX2, Hsp90a, Hsp90b, cdc25A, CDK2, RagC, CSF1R, EGFR, AKR1B1, JAK3, MNK2 and p38 were bound with all four kusunokinin stereoisomers. *Trans*-(−)-kusunokinin bound CSF1R with the highest specificity, whereas the other three forms bound AKR1B1. These findings suggested that when cells were treated with *trans*-(+)-kusunokinin, the *trans*-(−)-isoform had a higher chance of binding CSF1R than other candidates, whereas some *trans*-(−)-isoform and *trans*-(+)-kusunokinin may have crossed the cell membrane and inhibited other proteins such as AKR1B1. Figure 11 depicts a pathway containing the 12 proteins with the highest docking score and related molecules.

CSF1R, a tyrosine kinase transmembrane protein, acts as a receptor for CSF1 and IL-34 [13,15,16,17,18,58,59,60]. This protein promotes cancer cell proliferation, invasion and metastasis via the MEK/ERK or PI3K/AKT pathways [16,19,20,21,61]. CSF1R mediates PI3K/AKT leading to the activation of cMyc and upregulates cyclins and CDKs driving the cell cycle and proliferation [20,61]. Activation of MEK/ERK by CSF1R upregulates COX2 and PI3K/AKT, resulting in activation of anti-apoptosis and survival [15,19,20]. Moreover, CSF1R-intermediated signaling molecules are linked to other pathways; for example, RagC [62] and EGFR-mediated MEK/ERK cascade [52,53,54,63]. COX2 also activates AKR1B1 and drives metastasis [64].

Considering the docking score alone was insufficient for target protein selection (Table 1) due to that some proteins did not differ in binding energies among the four kusunokinin isoforms. Therefore, we presented a novel sorting method for drug-target screening. The docking score of kusunokinin isomers was recalculated as a rank score from 4 to 1 based on the binding possibility of the isomer. The rank score was then given a weighted score of 3-2-1 based on the protein’s likelihood of drug exposure. The weighted score was determined by target subcellular localization, which was an important factor in drug discovery [48,49,50,51,65]. Thus, we divided proteins into three groups and gave a score according to their role: membrane-bound proteins (3), cytosol proteins (2) and nuclear proteins (1). The membrane protein is the most suitable for drug targets that do not require membrane permeation [36,66,67,68]. Our scoring method facilitated protein selection for in vitro research.

The weighted sum score of *trans*-(−)-kusunokinin indicated that it was the most potent compound in the racemic mixture, exhibiting high selectivity in binding to proteins involved in cell growth, signaling molecules and anti-apoptosis (Table 2). The sum and weighted scores of *trans*-(−)-kusunokinin on CSF1R indicated that the *trans*-(−)-isoform was the active component in the racemic mixture, which accounts for the decision on CSF1R as the target of interest. Unlike *trans*-(±)-kusunokinin, the two *cis*-isomers exhibited a similar score on the weighted score on cell growth and anti-apoptotic proteins. Moreover, *cis*-(+)-kusunokinin isomers tend to be specific with metastasis proteins more than others. The action of *cis*-(+)-kusunokinin isomers should be verified by conducting additional research on the metastasis pathway in the near future.

Differences in configuration of kusunokinin stereoisomers affected binding poses and distinguishable binding energies. The variability of four kusunokinin binding energies within the same protein was investigated via one-way ANOVA (Appendix A). CSF1R had the highest selectivity toward *trans*-(−)-kusunokinin, whereas EGFR had the smallest difference between the docking score of the four isomers. Although the variances of EGFR, MNK2, JAK3 and AKR1B1 were all within the same range, EGFR was selected for binding pocket comparison to CSF1R because it is a membrane-bound protein from the proliferation and cell growth group, like CSF1R. Both CSF1R and EGFR have “glove-shaped” pockets; however, free binding of stereoisomers was allowed due to wide EGFR’s allosteric site and lack of aromatic residues. The stereoselectivity of CSF1R resulted from the pocket shape, which required a planar arrangement of the ligand to fit the pocket. The proteins of all isomers with good binding energies had spherical binding pockets, allowing isomers to bend into their preferred pose with less restriction than the narrow pocket in CSF1R.

The number and position of aromatic residues within the binding site also contribute to kusunokinin isoform stereoselectivity. The bond rotation of the kusunokinin aromatic rings had little effect on the binding energies due to the small number of aromatic residues. The proteins lacking isomeric selectivity relied primarily on hydrogen bonds and hydrophobic interactions. The differences in docking score among isomers with spherical protein pockets were caused by the different binding poses, which formed different hydrogen bonds between the ligand’s methoxy groups and appropriate residues within the binding site.

All four isomers bound CSF1R at the allosteric pocket, but they were unable to stretch their formaldehyde ethylene acetal rings into the ATP binding area like pexidartinib’s pyrimidine rings. The ligands did, however, bind CSF1R at the comparatively binding site to the crystal structure of CSF1R-pexidartinib, hinting at the possible inhibitory effects. Similar to pexidartinib, *trans*-(−)- and *cis*-(−)-isomers formed π-π stacking with Trp550 in JM [42,69]. *Trans*-(+)- and *cis*-(+)-isomers, on the other hand, could not fit into the JM region and were thus forced out to form the π-π interaction with the outer residues (Phe797 or Tyr665, respectively). Phe797 at the start of the activation loop (AL) in DFG motifs indicated whether CSF1R was activated (DFG-out) or inactivated (DFG-in) [70]. The π-π interaction of *trans*-(+)-kusunokinin’s benzene rings with Phe797 could lead to an inhibition of phosphorylation of the DFG motif and thus prevent CSF1R activation. However, the interaction of *trans*-(+)-kusunokinin with CSF1R was found to be less stable than that of *trans*-(−)-kusunokinin. MD simulation also illustrated that, in the JM region, the π-π interaction between the benzene rings of *trans*-(−)-kusunokinin and Trp550 stabilized the CSF1R conformation, which was a critical structure for improving potency and/or selectivity towards CSF1R. However, *trans*-(−)-kusunokinin, *trans*-(+)-kusunokinin and pexidartinib caused distinct conformational changes in CSF1R. An in vitro study is required to confirm *trans*-(±)-kusunokinin-CSF1R binding.

Lower capital costs and/or difficulty in chiral synthesis rationalized the usable racemic drug form over a single active enantiomer [56,57,71,72]. Pure enantiomers are usually expensive to chemically synthesize on a commercial scale [72,73]. As a result, the majority of available drugs are marketed as racemates, which are equimolar mixtures of two enantiomers, such as ibuprofen [74]. The main target protein, CSF1R, demonstrated strong selectivity toward the *trans*-(−) isoform and at the same time forced the *trans*-(+) isoform out of the pocket, as the MD simulation over time suggested. Therefore, *trans*-(±)-kusunokinin appeared to be a promising selective CSF1R racemic inhibitor. These statements validated the use of racemic *trans*-(±)-kusunokinin as a potential candidate or developed scaffold for targeting CSF1R and other proteins in the CSF1R breast cancer progression pathway.

## 4. Materials and Methods

### 4.1. Molecular Docking

Molecular docking was performed on 57 breast cancer-associated proteins using AutoDock4 version 4.2 [75]. The four kusunokinin stereoisomers were chosen as the docked ligands: *trans*-(−)-kusunokinin, *trans*-(+)-kusunokinin, *cis*-(+)-isomer and *cis*-(−)-isomer.

#### 4.1.1. Target Protein Selection and Preparation

The candidate protein was selected based on breast cancer-CSF1R associated pathways, then grouped as anti-apoptosis and survival, cell growth and proliferation, metastasis, or signal molecules. The protein crystal structure, in a PDB format, was acquired from the Protein Data Bank (https://www.rcsb.org, accessed on 19 August 2021). The monomeric form was chosen with co-crystallized ligand(s), solvent molecules and/or other non-protein molecules removed. Only cofactor molecules such as Zn^2+^ and Mg^2+^ were retained. The structure preparation was performed using Visual Molecular Dynamics (VMD) [76]. All polar hydrogens were added corresponding to the protonation state at pH 7 using Autodock Auxiliary Tool (ADT) version 1.5.6.

#### 4.1.2. Ligand Preparation

The 3D structure of *trans*-(−)-kusunokinin (Pubchem CID: 384876), *trans*-(+)-kusunokinin (Pubchem CID: 92154432), as well as each target protein inhibitor, were retrieved from PubChem (https://pubchem.ncbi.nlm.nih.gov, accessed on 19 August 2021). All retrieved files were in SDF format, and the files were then converted into PDB format via the Online SMILE Translator and Structure File Generator (https://cactus.nci.nih.gov/translate/index.html, accessed on 19 August 2021). For the *cis*-isomers, both structures were optimized in Open Babel software using a steepest descent algorithm with a convergence of 10^−10^ kcal/mol. The ligand structure was finally changed into a PDBQT file format using ADT prior to molecular docking.

#### 4.1.3. Docking Procedure and Binding Energy Calculation

The x-y-z grid point numbers of 120-120-120 with the center at the macromolecule were exploited in the docking study. Since we performed binding docking assumption, these chosen x-y-z grid point numbers had covered the protein receptor, ensuring that randomization was done without bias. The grid spacing was set as 0.375 Å. A genetic algorithm (GA) with 50 runs and a population size of 200 was operated with default parameters in AutoDock4. A Lamarckian genetic algorithm 4.2 was applied to predict bound conformations. The conformation with the lowest docking score in kcal/mol was considered as the best pose. The docking score of kusunokinin stereoisomers and its inhibitor to each target protein were reported and then assessed by the scoring system. 

#### 4.1.4. Target Protein Selection for Ligand-Protein Interaction Analysis

The docking score of the target protein known inhibitor, as well as the character of protein binding pocket and the interacted amino acid residues, were considered in this sorting step in order to reduce the number of candidate proteins to the most promising kusunokinin-derived drug targets for future study. A target(s)-choosing criteria for post-docking analysis required that the protein had a lower docking score of kusunokinin isomer than those of known inhibitor and the binding pocket of the said kusunokinin isomer and the known were similar. These criteria are based on a hypothesis that the same binding site and a similar docked pose between the ligand of interest and the known inhibitor could imply the inhibitory effect of the ligand to that protein.

#### 4.1.5. Statistical Analysis

The relative binding energies from the four kusunokinin isoforms within the same protein were assessed by the one-way ANOVA using the Calc package in Ubuntu. *p* < 0.01 was considered to indicate a statistically significant difference among isomer docking scores.

### 4.2. Scoring Procedure

The scoring procedure was carried out to investigate the preferred form of kusunokinin and its potential target proteins, which could be the cause of the contradictory effects observed in previous studies. The docking score of each kusunokinin stereoisomeric forms and the individual proteins was re-categorized as a rank score in 4-3-2-1 manner from the highest binding possibility (4) to the lowest binding possibility (1).

Weight scores ranging from 3 for transmembrane proteins, 2 for cytoplasmic protein and 1 for nuclear protein, based on drug-accessibility, were applied to the rank score as a multiplier to emphasize the differences in each kusunokinin isoform’s binding ability. This criterion was earlier illustrated in Figure 2.

The summations of the rank score and weighted rank score within each protein group were computed and presented as the sum score and weighted sum score, respectively. The weighted sum score indicated the preferred-kusunokinin form and the protein group which most likely to be the target of the kusunokinin most-preferred isoform.

### 4.3. Molecular Dynamics (MD) Simulation

#### 4.3.1. System Preparation of the Compound-Protein Complex

AMBER20 was used for molecular dynamics (MD) simulation to simulate the dynamic aqueous condition as a function of temperature and pressure. The docked PDB structure of the kusunokinin stereoisomer-protein complex was used as a starting point. All of the previous polar hydrogen from AutoDock4 was removed and full hydrogen atoms were added using the AMBER20 package’s LeAP module [77]. At pH 7, the corresponding state is all ionizable amino acid side chains. If necessary, the disulfide bond(s) were introduced. TIP3P water solvated the complex at a distance of 14 from the protein surface, resulting in approximately 17,400 water molecules. Na^+^ or Cl^−^ were used to neutralize the solvated system. To simulate a 0.15 M NaCl solution, 32 NaCl pairs were used.

#### 4.3.2. Simulation Protocol of the Compound-Protein Complex

The AMBER20 force field was used to parameterize the protein and solvent. The previous study’s kusunokinin RESP charge was used [42]. The protein solution system was energy-minimized for 1000 steps in a periodic boundary condition using steepest descent and conjugate gradient methods. The energy-minimized structure was then equilibrated at 298 K (25 °C) using Langevin Dynamics, also known as the NVT ensemble. Each NVT ensemble lasted 200 ps and had a time step of 1 fs. The system was then placed into an isothermal and isobaric (NPT) ensemble using the Berendsen algorithm [78] for 300 ns at 298 K and 1.013 bar (1 atm) with a time step of 2 fs. The cut-off value of 12 was used to handle non-bonded interactions. The electrostatic interaction was calculated using the PME summation. All H-X bonds were restrained using the SHAKE technique.

Finally, a snapshot MD trajectory of 15,000 equidistant points was used. The first 10,000 snapshots from the 0 to 200 ns MD trajectory were omitted and the last 100 ns of the NPT trajectory was used to calculate the conformational parameters and relative binding energy. VMD was used to visualize the structure and analyze the binding pocket. The RMSD plot acquisition was calculated from VMD and visualized using the Grace package in Ubuntu.

#### 4.3.3. Relative Binding Free Energy Evaluation of the Compound-Protein Complex

The average relative binding free energies (ΔG) in kcal/mol were calculated from MD trajectories using molecular mechanics/generalized Born-surface area (MM/GBSA) and molecular mechanics/Poisson-Boltzmann surface area (MM/PBSA) [78]. The computation protocol was summarized in the previous study [79].

## 5. Conclusions

The molecular simulation and our newly proposed ranking system predicted that *trans*-(−)-kusunokinin targeted CSF1R, whereas *trans*-(+)-kusunokinin and *cis*-(±)-isomer inhibited AKR1B1. In the cell growth and proliferation group, *trans*-(−)-kusunokinin was the most potent form targeting proteins. The possibility of *cis*-(±)-isomer targeting proteins in the metastasis group was to be further investigated. Because of the narrow binding pocket and the aromatic Trp550, the stereoisomeric preference of *trans*-(−)-kusunokinin was observed in CSF1R via selective π-π interaction with a planar arranged *trans*-(−)-kusunokinin. Other targets, such as EGFR binding pockets, were larger than those of CSF1R and lacked aromatic residues, allowing kusunokinin isomers to bend and bind freely. Our study concluded that the racemic *trans*-(±)-kusunokinin may have inhibited breast cancer cells primarily through the binding and suppression of CSF1R by *trans*-(−)-kusunokinin, in addition to the binding of *trans*-(−) or *trans*-(+) isomers to other target proteins.

## Figures and Tables

**Figure 1 molecules-27-04194-f001:**
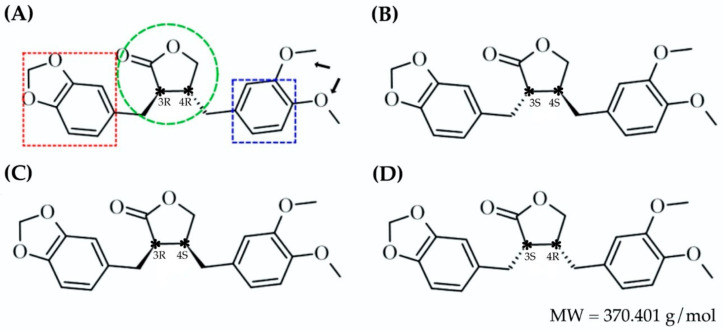
Enantiomeric forms of kusunokinin: (**A**) *trans*-(−)-kusunokinin; (**B**) *trans*-(+)- isomer; (**C**) *cis*-(−)-isomer; (**D**) *cis*-(+)-isomer. The green circle, red box, blue box and black arrows represented γ-Butyrolactone, formaldehyde ethylene acetal, benzene ring and methoxy group, respectively. The absolute configuration of both chiral carbons was also addressed. The star sign (*) represented chiral carbon atoms of each kusunokinin isomers.

**Figure 2 molecules-27-04194-f002:**
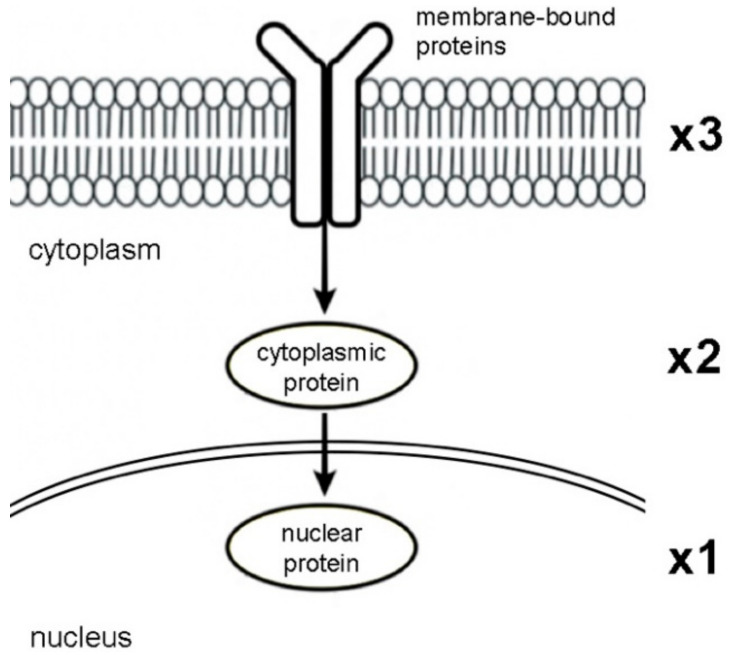
Weighting score of ligand-accessibility of the protein. The score 3, 2 and 1 represented transmembrane, cytoplasmic and nuclear proteins, respectively.

**Figure 3 molecules-27-04194-f003:**
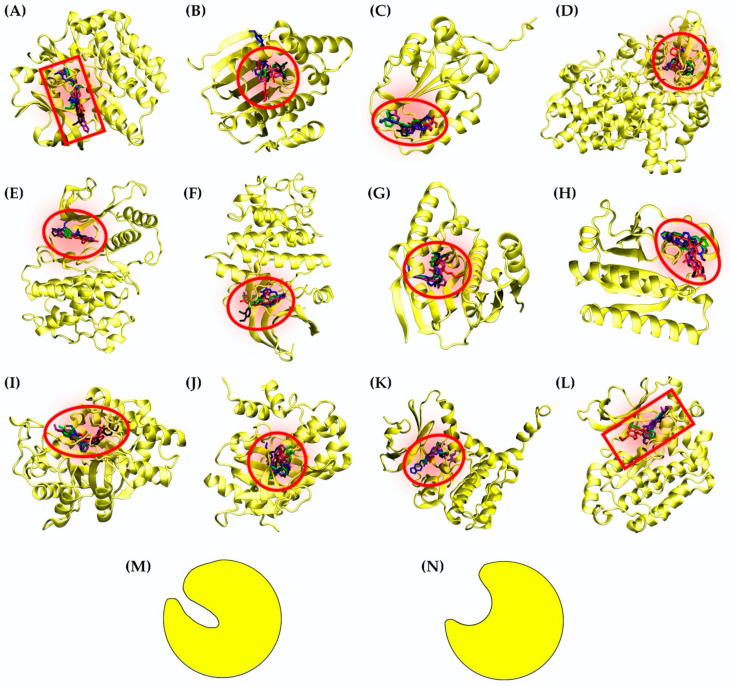
Kusunokinin alignment in the 12 possible targets’ binding pockets: (**A**) CSF1R; (**B**) Hsp90b; (**C**) cdc25A; (**D**) COX2; (**E**) p38; (**F**) CDK2; (**G**) Hsp90a; (**H**) RagC; (**I**) AKR1B1; (**J**) JAK3; (**K**) MNK2; (**L**) EGFR; (**M**) glove-like structure; (**N**) spherical structure. Blue, black, green and purple sticks represented the docking poses of *trans*-(−)-kusunokinin, *trans*-(+)-kusunokinin, *cis*-(−)-isomer and *cis*-(+)-isomer, respectively. The red rectangle represented the glove-like binding pocket. The red circle represented the spherical binding pocket.

**Figure 4 molecules-27-04194-f004:**
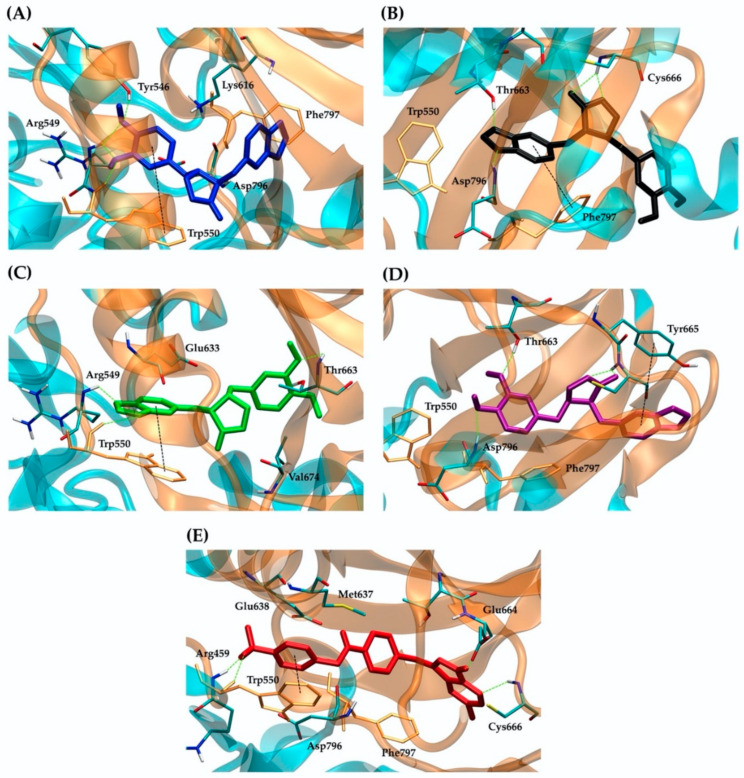
Binding interaction of ligands with a CSF1R kinase structure: (**A**) *trans*-(−)-kusunokinin (blue); (**B**) *trans*-(+)-kusunokinin (black); (**C**) *cis*-(−)-isomer (green); (**D**) *cis*-(+)-isomer (purple); (**E**) pexidartinib (red). The green dash lines represented hydrogen bonding. Trp550, Tyr665 and Phe797 were responsible for π-π interaction (black dash lines). Docking poses of the ligands and amino acids interacting with the ligands through hydrophobic interaction were shown as sticks.

**Figure 5 molecules-27-04194-f005:**
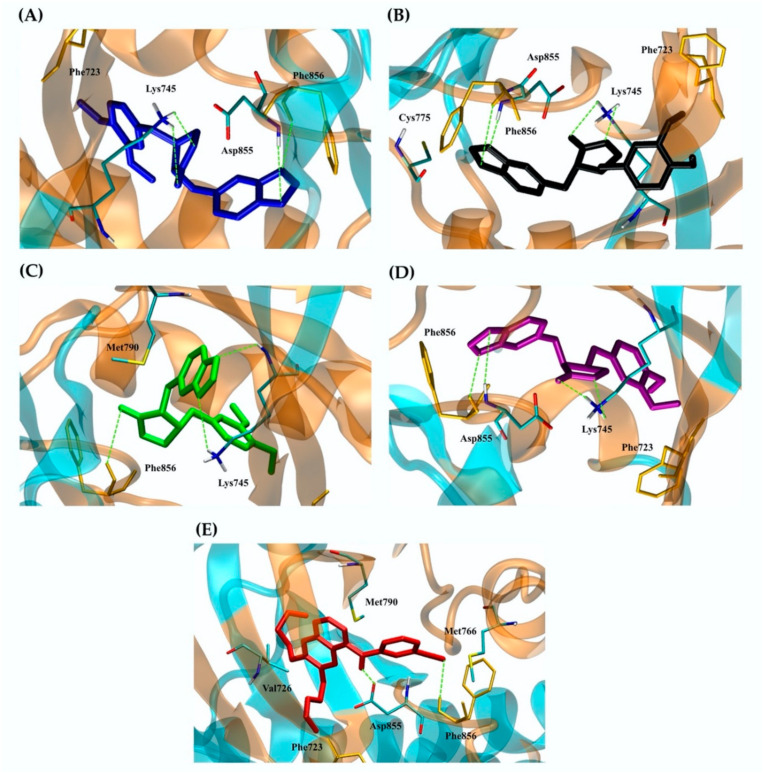
Binding interaction of ligands with an EGFR structure: (**A**) *trans*-(−)-kusunokinin (blue); (**B**) *trans*-(+)-kusunokinin (black); (**C**) *cis*-(−)-isomer (green); (**D**) *cis*-(+)-isomer (purple); (**E**) erlotinib (red). The green dash lines represented hydrogen bonding. Docking poses of the ligands and amino acids interacting with the ligands through hydrophobic interaction or π-sulfur interaction were shown as sticks.

**Figure 6 molecules-27-04194-f006:**
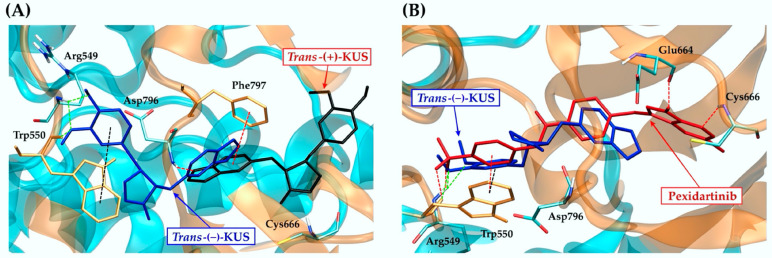
CSF1R binding pocket alignment: (**A**) *trans*-(−)-kusunokinin (blue) compared to *trans*-(+)-kusunokinin (black); (**B**) *trans*-(−)-kusunokinin (blue) compared to pexidartinib (red). Arg549 and Trp550 were responsible for hydrogen bonding between *trans*-(−)-kusunokinin and CSF1R (green dash line). Trp550 was responsible for π-π interaction between *trans*-(−)-kusunokinin and CSF1R (black dash lines). The red dash line represented hydrogen bonding or π-π stacking of the *trans*-(+)-kusunokinin or known inhibitor. Docking poses of the ligands and amino acids interacting with the ligands through hydrophobic interaction were shown as sticks.

**Figure 7 molecules-27-04194-f007:**
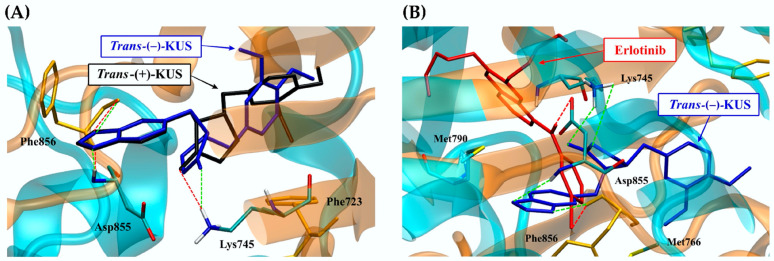
EGFR binding pocket alignment: (**A**) *trans*-(−)-kusunokinin (blue) compared to *trans*-(+)-kusunokinin (black); (**B**) *trans*-(−)-kusunokinin (blue) compared to erlotinib (red). Lys745, Asp855 and Phe856 were responsible for hydrogen bonding between *trans*-(−)-kusunokinin and EGFR (green dash line). The red dash line represented hydrogen bonding of *trans*-(+)-kusunokinin or the known inhibitor. Docking poses of the ligands and amino acids interacting with the ligands through hydrophobic interaction or π-sulfur were shown as sticks.

**Figure 8 molecules-27-04194-f008:**
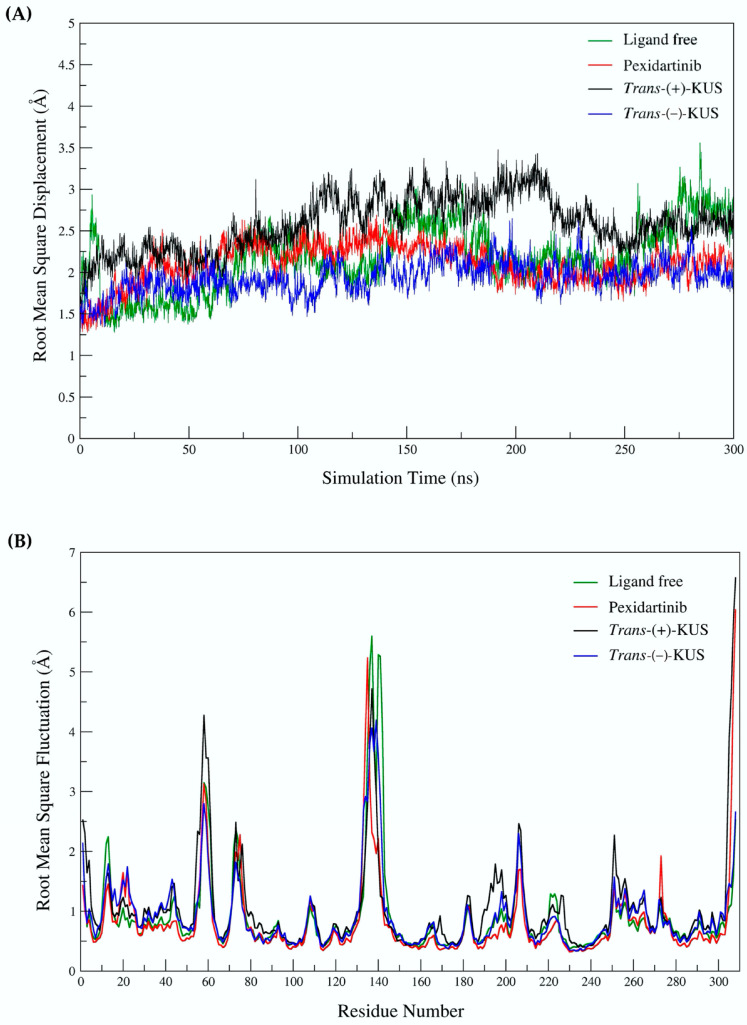
Structural parameters obtained from MD simulation: (**A**) root-mean square displacement (RMSD); (**B**) root-mean square fluctuation (RMSF). Both RMSD and RMSF were reported in Angstrom.

**Figure 9 molecules-27-04194-f009:**
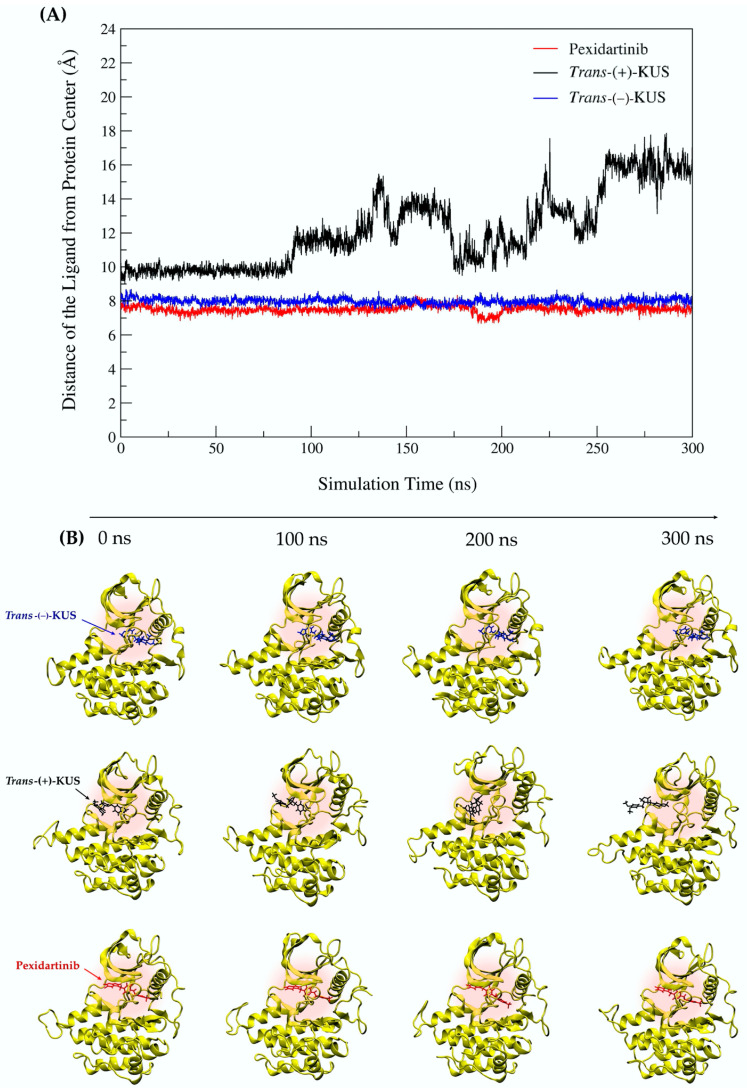
Ligand binding overtime: (**A**) distance of the ligands from protein center; (**B**) the position of ligand in the ligand-protein complex at the starting point (0 ns), 100 ns, 200 ns and 300 ns. The CSF1R structure was represented in yellow. The *trans*-(−)-kusunokinin, *trans*-(+)-kusunokinin and pexidartinib were represented in blue, black and red, respectively.

**Figure 10 molecules-27-04194-f010:**
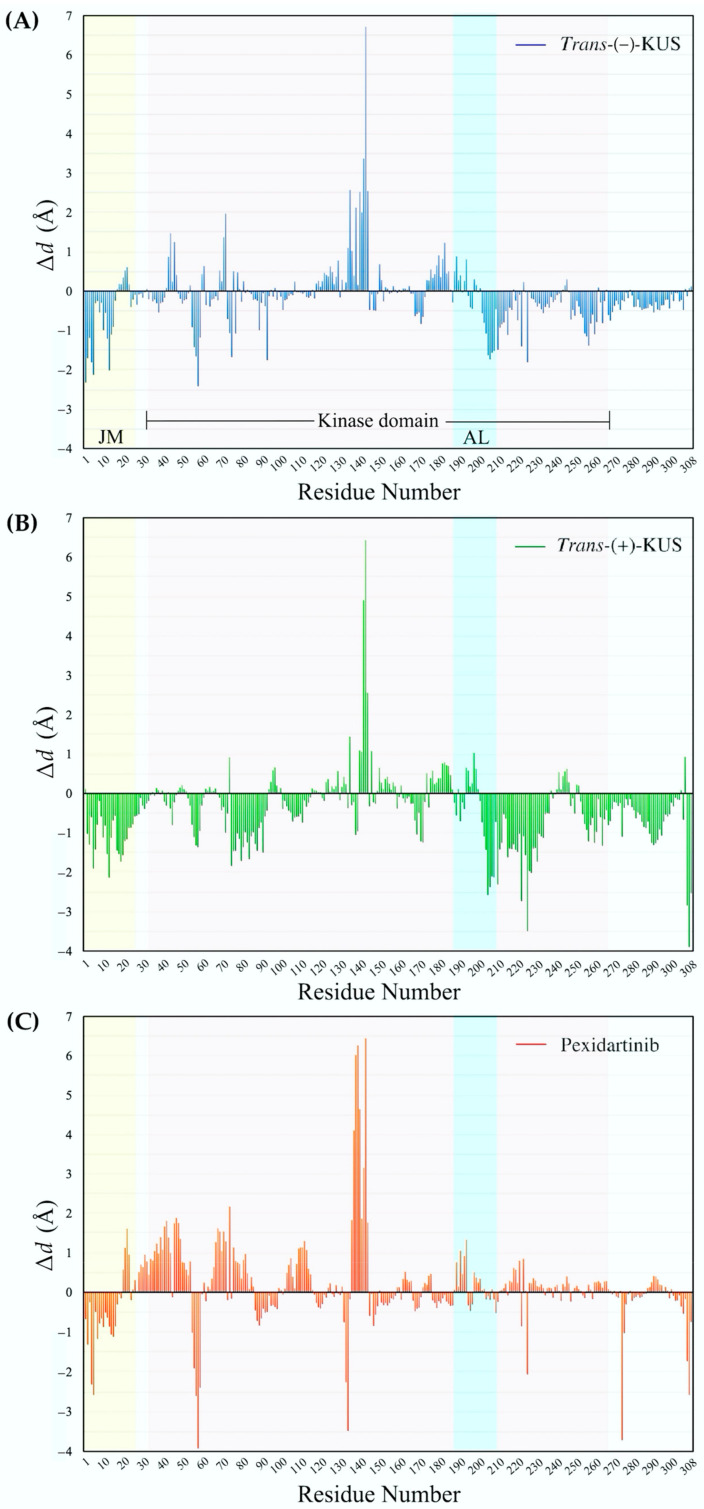
Pattern of distance difference of ligand-bound CSF1R with respect to the ligand-free state: (**A**) distance pattern of *trans*-(−)-kusunokinin-bound CSF1R; (**B**) *trans*-(+)-kusunokinin-bound CSF1R; (**C**) pexidartinib-bound CSF1R, aligned with the CSF1R-unbound structure.

**Figure 11 molecules-27-04194-f011:**
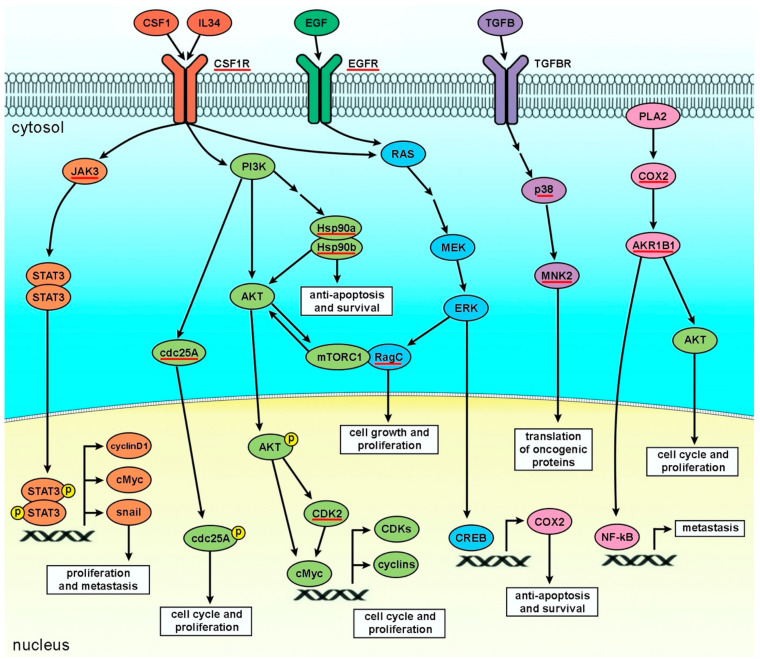
Top 12 candidate proteins of four kusunokinin isomers (red underlines) in the breast cancer progression pathway. *Trans*-(−)-kusunokinin showed best binding to CSF1R while *trans*-(+)-kusunokinin and the two *cis*-isomers possibly targeted AKR1B1.

**Table 1 molecules-27-04194-t001:** The docking score of four possible kusunokinin isomers and inhibitors with 60 candidate proteins.

Target Name	PDB Code	Kusunokinin Isomer Docking Score ^1^				Known Inhibitor	
		*Trans*-(−)	*Trans*-*(+)*	*Cis*-(−)	*Cis*-*(+)*	Name	Docking Score
**Anti-apoptosis and Survival**							
COX2	5IKR	−10.60	−9.68	−9.62	−10.57	Sorafenib	−9.06
Hsp70	3ATV	−8.93	−7.84	−8.43	−9.03	Apoptozole	−8.24
Hsp90a	3O0I	−10.60	−10.09	−10.43	−10.77	Pu-H54	−9.43
Hsp90b	3NMQ	−10.80	−9.28	−10.63	−9.74	EC44	−10.20
PAK6	4KS8	−8.55	−7.56	−8.09	−8.39	Sunitinib	−7.50
Survivin	1F3H	−8.78	−8.63	−8.76	−8.13	YM155	−7.25
XIAP	5OQW	−7.67	−6.74	−7.70	−7.49	A4E	−10.66
**Cell Growth and Proliferation**							
cMyc	6G6J	−7.30	−6.32	−7.22	−6.55	MYCi975	−6.19
CDK2	4J52	−9.74	−9.14	−9.90	−9.90	Dinaciclib	−8.93
PLK1	4OYA	−8.87	−8.87	−8.92	−8.84	Pyrimidodiazepinone	−5.92
ADC10	1C25	−12.59	−11.32	−12.21	−12.12	SQ-22536	−8.23
AR	1CWT	−8.50	−8.35	−9.36	−9.03	(*R*)-Bicalutamide	−8.55
cdc25A	4Y72	−9.19	−7.14	−7.98	−8.97	Quinonoid	−6.90
cdc25B	1FVV	−7.62	−7.46	−7.63	−7.78	Quinonoid	−6.91
CDK1	6P8E	−10.34	−9.27	−9.79	−9.69	CGP74514A	−10.96
CDK4	1FVV	−8.16	−7.22	−7.32	−7.55	Palbociclib	−9.03
cyclinA	4Y72	−8.89	−9.56	−8.92	−8.69	ligand 107	−9.76
cyclinB1	6P8E	−10.45	−10.08	−10.24	−10.84	Q27097368	−9.40
cyclinD1	1L09	−8.18	−7.04	−8.20	−8.42	Fascaplysin	−7.57
ER	1SJ0	−8.96	−8.43	−9.02	−8.87	E4D	−11.74
GSK3b	4JSV	−10.64	−10.01	−9.91	−10.05	Tideglusib	−8.59
mTOR	4UJA	−9.12	−8.46	−9.09	−8.95	AZD8055	−10.21
PKA	3IW4	−9.48	−8.46	−9.58	−9.61	AT13148	−12.83
PKC	3LLU	−9.83	−8.99	−9.87	−9.79	Enzastaurin	−12.69
RagC	4R7H	−9.21	−8.73	−8.95	−8.82	Palomid-529	−8.85
CSF1R	4HJO	−11.84	−9.29	−10.53	−10.40	Pexidartinib	−11.59
EGFR	5 × 02	−9.71	−6.45	−9.69	−7.14	Erlotinib	−8.82
FLT3	1N26	−9.24	−8.76	−9.11	−9.13	Gilteritinib	−9.64
IL6R	6G6J	−7.74	−7.48	−7.61	−7.73	Terminolic acid	−7.11
**Metastasis**							
cFos	1FOS	−5.63	−5.07	−5.32	−5.51	T-5224	−6.20
cJun	1FOS	−5.66	−4.95	−5.27	−5.51	T-5224	−7.72
mmp12	4XCT	−11.19	−10.23	−10.71	−10.80	BAY-7598	−12.47
mmp9	5LAB	−10.41	−10.48	−10.37	−9.64	ARP101	−12.27
snail	3W5K	−7.85	−7.53	−7.49	−7.47	Chembl4517265	−7.70
AKR1B1	4JIR	−11.32	−11.15	−11.46	−11.59	Epalrestat	−10.14
ALP	2GLQ	−7.26	−7.95	−7.36	−7.34	Levamisole	−6.08
PALP	3MK0	−6.92	−6.77	−7.69	−7.48	Levamisole	−5.79
TGFBR1	1E3G	−7.08	−6.92	−7.06	−7.62	SB431542	−7.17
TGFBR2	2PJY	−7.96	−7.73	−7.88	−8.21	J2V	−7.10
**Signaling molecules**							
AKT	1GZN	−9.64	−8.97	−9.12	−9.23	Capivasertib	−8.50
CRAF	3OMV	−8.60	−8.11	−9.14	−8.82	Sorafenib	−9.06
ERK1	4QTB	−9.63	−9.24	−9.54	−9.65	SCH772984	−12.96
ERK2	5BUJ	−9.15	−8.65	−9.11	−9.42	Q27455064	−8.39
Grb2	1GRI	−8.03	−7.32	−7.78	−7.92	CGP-78850	−7.88
IKK	4KIK	−7.89	−7.70	−7.80	−8.18	Dehydrocostus Lactone	−7.28
JAK1	5HX8	−8.64	−8.56	−8.60	−8.44	66P	−10.52
JAK2	3KRR	−8.20	−7.66	−8.40	−8.17	NVP-BSK805	−11.22
JAK3	5TTV	−8.97	−8.62	−9.03	−8.81	Inhibitor 6	−7.30
K-RAS	4M1W	−7.55	−7.19	−7.85	−7.69	Sotorasib	−7.11
MEK1	2P55	−10.34	−9.48	−10.63	−10.14	Trametinib	−13.05
MEK2	1S9I	−10.17	−9.41	−10.95	−10.06	Trametinib	−12.17
MNK2	6JLR	−9.86	−9.60	−10.02	−9.94	BV9	−8.64
p38	4MYG	−8.73	−7.89	−8.63	−8.61	PD169316	−7.94
PI3K	3CSF	−9.47	−8.66	−9.12	−9.06	Apitolisib	−9.55
PIM1	1YXV	−9.38	−8.03	−9.16	−9.30	6SD	−6.95
RAC1	1HH4	−10.12	−9.33	−9.36	−10.12	NSC-23766	−6.94
SMAD3	1MK2	−8.33	−7.78	−8.06	−7.91	SIS3	−9.05
STAT1	1YVL	−7.41	−7.75	−7.49	−7.37	Fludarabine	−5.41
STAT3	6TLC	−7.84	−7.51	−7.63	−7.82	STAT3-IN-3	−9.87
STAT5	6MBZ	−9.52	−9.27	−8.78	−8.96	IN-2	−8.45

^1^ The docking score from redocking of target proteins and native ligands of PDB entries was represented from the predicted relative binding energy in kcal/mol.

**Table 2 molecules-27-04194-t002:** Candidate proteins with rank score and weight score.

Target Name	Weight ^1^	Kusunokinin Isomer Rank Score ^2^							
		*Trans*-(−)		*Trans*-(+)		*Cis*-(−)		*Cis*-(+)	
**Anti-apoptosis and Survival**									
COX2	2	4	(8)	2	(4)	1	(2)	3	(6)
Hsp70	2	3	(6)	1	(2)	2	(4)	4	(8)
Hsp90a	2	3	(6)	1	(2)	2	(4)	4	(8)
Hsp90b	2	4	(8)	1	(2)	3	(6)	2	(4)
PAK6	2	4	(8)	1	(2)	2	(4)	3	(6)
Surviving	2	4	(8)	2	(4)	3	(6)	1	(2)
XIAP	2	3	(6)	1	(2)	4	(8)	2	(4)
**Sum score (weighted) ^3^**		21		(54)		8		(19)	
**Cell growth and Proliferation**									
cMyc	1	4	(4)	1	(1)	3	(3)	2	(2)
CDK2	1	2	(2)	1	(1)	4	(4)	4	(4)
PLK1	1	3	(3)	3	(3)	4	(4)	1	(1)
ADC10	2	4	(8)	1	(2)	3	(6)	2	(4)
AR	2	2	(4)	1	(2)	4	(8)	3	(6)
cdc25A	2	4	(8)	1	(2)	2	(4)	3	(6)
cdc25B	2	2	(4)	1	(2)	3	(6)	4	(8)
CDK1	2	4	(8)	1	(2)	3	(6)	2	(4)
CDK4	2	4	(8)	1	(2)	2	(4)	3	(6)
cyclinA	2	2	(4)	4	(8)	3	(6)	1	(2)
cyclinB1	2	3	(6)	1	(2)	2	(4)	4	(8)
cyclinD1	2	2	(4)	1	(2)	3	(6)	4	(8)
ER	2	3	(6)	1	(2)	4	(8)	2	(4)
GSK3b	2	4	(8)	2	(4)	1	(2)	3	(6)
mTOR	2	4	(8)	1	(2)	3	(6)	2	(4)
PKA	2	2	(4)	1	(2)	3	(6)	4	(8)
PKC	2	3	(6)	1	(2)	4	(8)	2	(4)
RagC	2	4	(8)	1	(2)	3	(6)	2	(4)
CSF1R	3	4	(12)	1	(3)	2	(6)	3	(9)
EGFR	3	4	(12)	1	(3)	3	(9)	2	(6)
FLT3	3	4	(12)	1	(3)	2	(6)	3	(9)
IL6R	3	4	(12)	1	(3)	2	(6)	3	(9)
**Sum score (weighted)**		68	(151)	25	(55)	48	(120)	62	(122)
**Metastasis**									
cFos	1	4	(4)	1	(1)	2	(2)	3	(3)
cJun	1	4	(4)	1	(1)	2	(2)	3	(3)
mmp12	1	4	(4)	1	(1)	2	(2)	3	(3)
mmp9	1	3	(3)	4	(4)	2	(2)	1	(1)
Snail	1	4	(4)	3	(3)	2	(2)	1	(1)
AKR1B1	2	2	(4)	1	(2)	3	(6)	4	(8)
ALP	2	1	(2)	4	(8)	3	(6)	2	(4)
PALP	2	2	(4)	1	(2)	4	(8)	3	(6)
TGFBR1	3	3	(9)	1	(3)	2	(6)	4	(12)
TGFBR2	3	3	(9)	1	(3)	2	(6)	4	(12)
**Sum score (weighted)**		33	(52)	19	(30)	28	(45)	29	(56)
**Signaling molecules**									
AKT	2	4	(8)	1	(2)	2	(4)	3	(6)
CRAF	2	2	(4)	1	(2)	4	(8)	3	(6)
ERK1	2	3	(6)	1	(2)	2	(4)	4	(8)
ERK2	2	3	(6)	1	(2)	2	(4)	4	(8)
Grb2	2	4	(8)	1	(2)	2	(4)	3	(6)
IKK	2	3	(6)	1	(2)	2	(4)	4	(8)
JAK1	2	4	(8)	2	(4)	3	(6)	1	(2)
JAK2	2	3	(6)	1	(2)	4	(8)	2	(4)
JAK3	2	3	(6)	1	(2)	4	(8)	2	(4)
K-RAS	2	2	(4)	1	(2)	4	(8)	3	(6)
MEK1	2	3	(6)	1	(2)	4	(8)	2	(4)
MEK2	2	3	(6)	1	(2)	4	(8)	2	(4)
MNK2	2	2	(4)	1	(2)	4	(8)	3	(6)
p38	2	2	(4)	4	(8)	3	(6)	1	(2)
PI3K	2	4	(8)	1	(2)	3	(6)	2	(4)
PIM1	2	4	(8)	1	(2)	2	(4)	3	(6)
RAC1	2	4	(8)	1	(2)	2	(4)	4	(8)
SMAD3	2	4	(8)	1	(2)	3	(6)	2	(4)
STAT1	2	2	(4)	4	(8)	3	(6)	1	(2)
STAT3	2	4	(8)	1	(2)	2	(4)	3	(6)
STAT5	2	4	(8)	3	(6)	1	(2)	2	(4)
**Sum score (weighted)**		71	(148)	57	(114)	61	(122)	32	(64)

^1^ Score for Ligand Accessibility (3 for membrane-bound proteins, which were most likely to contact with the drug, 2 for cytoplasm proteins and 1 for proteins inside nucleus which were the least exposed). ^2^ According to Table 1, the ranking score for kusunokinin stereoisomers is: 4 for the best binding ability prediction, 3 for the second, 2 for the third and 1 for the worst binding ability prediction. The blanket’s number represented the rank score multiplied by the protein weight score. ^3^ The sum score represents the total sum of the rank scores of each isoform within the same protein group. The blanket number represented the sum of rank scores multiplied by the weight score of each protein.

**Table 3 molecules-27-04194-t003:** Type of interaction between kusunokinin isomers/inhibitor with CSF1R and EGFR.

Target and Interaction	Compounds				
	*Trans*-(−)	*Trans*-(+)	*Cis*-(−)	*Cis*-(+)	Inhibitor ^1^
**CSF1R**					
π-π stacking	Trp550	Phe797	Trp550	Tyr665	Trp550
Hydrogen bond	Tyr546 Arg549 Trp550	Thr663 Cys666 Asp796	Arg549 Trp550 Thr663	Thr663 Tyr665 Asp796	Arg459 Trp550 Glu664 Cys666
**EGFR**					
Hydrogen bond	Lys745 Asp855 Phe856	Lys745 Asp855 Phe856	Lys745 Phe856	Lys745 Asp855 Phe856	Asp855 Phe856

^1^ Inhibitors of CSF1R and EGFR are pexidartinib and erlotinib, respectively.

**Table 4 molecules-27-04194-t004:** Relative binding free energies of ligand-CSF1R complexes evaluated from molecular dynamics simulations.

Ligand	MM/GBSA (kcal/mol)	MM/PBSA (kcal/mol)
*Trans*-(+)-kusunokinin	−22.00 ± 0.14	6.92 ± 0.14
*Trans*-(−)-kusunokinin	−54.67 ± 0.09	0.34 ± 0.15
Pexidartinib	−61.78 ± 0.09	−7.27 ± 0.14

## Data Availability

Not applicable.

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
