# Peer review of "Potential Stereoselective Binding of Trans-(±)-Kusunokinin and Cis-(±)-Kusunokinin Isomers to CSF1R"

_molecules, 2022, doi:10.3390/molecules27134194_

Round 1
Reviewer 1 Report
The authors describe an exhaustive computational study of the binding of Trans-(±)-Kusunokinin and Cis-(±) Kusunokinin isomers to CSF1R and other related proteins. The manuscript will be of interest to those working with computational tools such as docking and molecular dynamics, but specially to those working in anticancer drug design. However, there are some points to be considered.
1.- The title should be modified, in its actual form does not describes all the work performed. Additionally, the word “potential” should be included, for example, in its actual form the title would be “Potential stereoselectivity……..” because only in silico evidence is given, as is correctly used in the text.
2.- In the text the use of DG should be changed by is stated in table 1 “docking score”, it is the correct, because in docking simulations no DG is obtained.
3.- The syntaxes in line 168 should be modified because the target is the protein not the compound as is stated.
4.- Please include in materials and methods section the information corresponding to how the MD simulations for aggregation experiments were performed.
5.- Discussion section should be reduced to be more concrete about the most important findings, in its actual form contains a lot of unnecessary information.
Author Response
Review 1
Comments and Suggestions for Authors
- 1. The title should be modified, in its actual form does not describes all the work performed. Additionally, the word “potential” should be included, for example, in its actual form the title would be “Potential stereoselectivity……..” because only in silico evidence is given, as is correctly used in the text.
Response: We modified the title according to reviewer comment. “Potential Stereoselective Binding of Trans-(±)-Kusunokinin and Cis-(±)-Kusunokinin Isomers to CSF1R”
===================================================
2. In the text the use of ∆G should be changed by is stated in table 1 “docking score”, it is the correct, because in docking simulations no ∆G is obtained.
Response: The ∆G or ∆Gbind in the docking simulation context was all changed into “docking score” as suggested.
===================================================
3. The syntaxes in line 168 should be modified because the target is the protein not the compound as is stated.
Response: The sentence in line 168 was changed into “whereas cis-(+)-isomer was a potential candidate for the inhibition of metastasis proteins.” to respond this point.
===================================================
4. Please include in materials and methods section the information corresponding to how the MD simulations for aggregation experiments were performed.
Response: In our study, we performed the MD simulation containing one pair of respective kusunokinin enantiomers under the explicit 0.15 M NaCl solution. The simulation contained one pair of kusunokinin in the TIP3P water molecules. The other reviewer suggested that the aggregation could be formed via at least two of each enantiomer as the other reviewer suggested.
We agreed with the other reviewer that the simulation looked not realistic and could mislead the reader in case that the reader would cite this article in the future. We thus decided to remove this part from the manuscript. In our opinion, the aggregation is of minor point in the manuscript so without this part, the manuscript still provided the core information as the authors would like to.
===================================================
5.- Discussion section should be reduced to be more concrete about the most important findings, in its actual form contains a lot of unnecessary information.
Response: The discussion part was rewritten and made more concise as suggested.
===================================================

Reviewer 2 Report
In the manuscript titled “Stereoselective Binding of Trans-(±)-Kusunokinin and Cis-(±)-Kusunokinin Isomers to CSF1R” the authors report a computational study using docking and molecular dynamics (MD) on the natural product trans-(-)-kusunokokinin and on the other three possible stereoisomers, not yet investigated. The study is very interesting and cover many potential target, as well as the obtained results are very promising for further studies.
My comments are:
1. In Figure 1 please report the absolute stereochemistry of stereocenters (3R,4R) for trans-(-)-kusunokokinin, which is the known natural product, (3S,4S) for trans-(+)-kusunokokinin, (3R,4S) for cis-(-)- isomer and (3S, 4R) for cis-(+)- isomer, instead to use (+)/(-) at the chiral centers, which result ambiguous.
2. At row 72 it is reported:” Despite the fact that the cis-(±)-kusunokinin isomers have yet to be synthesized …”, but this is not completely correct, because Y. Moritani et al. in J. Org. Chem., 1996, 61, 6922-6930. DOI:10.1021/jo9601932, report the synthesis of a series of butyrolactone lignans, among them it is present the cis-isomer of kusunokinin (compound 2l in the cited article) as well as the trans isomer (compound 1l). Previously, in 1983, both isomers were isolated from Virola sebifera and fully characterised (both from a chemical and sterochemical point of view) by Lopes et al. (Phyrochemistry, 1983, 22,1516-1518, DOI: 10.1016/s0031 -9422(00)84055-8). Furthermore cis isomer was isolated and studied for its anti-estrogenic activity by Lee et al. in Arch. Pharm. Res., 2005, 28, 186-189. DOI:10.1007/bf02977713. In both the article cited above the molecule is not reported as cis-kusunokinin, but with the not IUPAC chemical name 3-(3,4-dimethoxybenzyl)-2-(3,4-methylenedioxybenzyl) butyrolactone (CAS Reg. N.: 88198-99-6 for (3R,4S) isomer and CAS Reg. N.:18884-25-8 for (3S, 4R) isomer).
3. Due to the known anti-estrogenic and alkaline phosphatese (APase) inhibition activities, as reported by Lee et al., the authors, to complete this study, should add a further docking calculation on these target.
4. In Table 1. the authors should insert, after the target column, a column containing the PDB code of the protein used in docking calculation, instead to put it in Table S1.
5. In order to visualize quickly the most important interactions, without to search in the text or in the figures, the authors should insert a further table reporting for each compound and for each target the type of interaction (H-bond, π-π interaction, charge-charge, and so on) and the amino acid residue involved in the interaction/s.
6. In paragraph 2.6 is reported the study by MD of potential aggregation in the racemic mixture of (±)-kusunokinin. It is not clear if the authors have performed this study in vacuo or in the presence of explicit solvent molecules. From Figure 11 (B-G) it seems that only one molecule of each kusunokinin enantiomer was used in the calculation. The possibility of aggregation could be between two molecule of the same enantiomer present in the racemic mixture, so the calculation on only one molecule of each enantiomer looks not realistic, and it needs to be performed on at least two molecules of each enantiomer or better on a more large numbers of molecules. In paragraph 4.3 a subparagraph, containing calculation details on the aggregation study, needs to be added.
7. In paragraph 4.1.3 the authors used a very large box centered on the entire target molecule of the protein, instead of centering a smaller box on the known binding site for each target protein, given the presence of the ligand in the X-ray structure. Note that in Autodock the dimensions of box are reported in “number of points” and the “grid spacing” should be set (normally by default it is 0.375 Å, which is a quarter of C-C bond length, whereas in Autodock Vina the grid spacing should be set at 1Å showing the size directly in Å). The values 126x126x126 reported in Å (in the experimental part) are not realistic because, the box appears much larger than the target protein. W. P. Feinstein and M. Brylinski report in the article titled “Calculating an optimal box size for ligand docking and virtual screening against experimental and predicted binding pockets” (Journal of Cheminformatics (2015) 7:18. DOI 10.1186/s13321-015-0067-5) a script for calculating an optimal box size. The use of such a large box, in addition to providing less accurate docking results, can insert the ligands at different sites than those that bind the reference ligand present in the X-ray structure. Usually, it is used a box which covers all the protein in the blind docking when the binding site of the ligand is unknown. Why the authors have chosen to use a very large box?
In conclusion, the manuscript can be accepted after making the corrections by following the indications given above in points 1-7.

Author Response
Reviewer 2
In the manuscript titled “Stereoselective Binding of Trans-(±)-Kusunokinin and Cis-(±)-Kusunokinin Isomers to CSF1R” the authors report a computational study using docking and molecular dynamics (MD) on the natural product trans-(-)-kusunokokinin and on the other three possible stereoisomers, not yet investigated. The study is very interesting and cover many potential target, as well as the obtained results are very promising for further studies.
My comments are:
1.In Figure 1 please report the absolute stereochemistry of stereocenters (3R,4R) for trans-(-)-kusunokokinin, which is the known natural product, (3S,4S) for trans-(+)-kusunokokinin, (3R,4S) for cis-(-)- isomer and (3S, 4R) for cis-(+)- isomer, instead to use (+)/(-) at the chiral centers, which result ambiguous.
Response: The Figure 1 was corrected as the reviewer suggested.
===================================================
2.At row 72 it is reported:” Despite the fact that the cis-(±)-kusunokinin isomers have yet to be synthesized …”, but this is not completely correct, because Y. Moritani et al.in J. Org. Chem., 1996, 61, 6922-6930. DOI:10.1021/jo9601932, report the synthesis of a series of butyrolactone lignans, among them it is present the cis-isomer of kusunokinin (compound 2l in the cited article) as well as the trans isomer (compound 1l). Previously, in 1983, both isomers were isolated from Virola sebifera and fully characterised (both from a chemical and sterochemical point of view) by Lopes et al. (Phyrochemistry, 1983, 22,1516-1518, DOI: 10.1016/s0031 -9422(00)84055-8). Furthermore cis isomer was isolated and studied for its anti-estrogenic activity by Lee et al. in Arch. Pharm. Res., 2005, 28, 186-189. DOI:10.1007/bf02977713. In both the article cited above the molecule is not reported as cis-kusunokinin, but with the not IUPAC chemical name 3-(3,4-dimethoxybenzyl)-2-(3,4-methylenedioxybenzyl) butyrolactone (CAS Reg. N.: 88198-99-6 for (3R,4S) isomer and CAS Reg. N.:18884-25-8 for (3S, 4R) isomer).
Response: In the revised manuscript, this raised issue was corrected as suggested. In line 57-58 and 62-63, the IUPAC names of the trans-isomers were added. Also in line 72-79, the IUPAC names of cis-isomers along with the context of suggested comments were also included in the introduction completely. We are really grateful for this point as we truly do not know this. Thanks very much.
===================================================
3.Due to the known anti-estrogenic and alkaline phosphatese (APase) inhibition activities, as reported by Lee et al., the authors, to complete this study, should add a further docking calculation on these targets.
Response: We have performed the molecular docking of kusunokinin isomers with estrogen receptor (ER) and alkaline phosphatase (AP) both human ALP and placental ALP in the Table 1, Table 2, Table S1 and Table S2.
===================================================
4.In Table 1. the authors should insert, after the target column, a column containing the PDB code of the protein used in docking calculation, instead to put it in Table S1.
Response: The PDB code of the protein target was inserted in the Table 1 as suggested.
===================================================
5.In order to visualize quickly the most important interactions, without to search in the text or in the figures, the authors should insert a further table reporting for each compound and for each target the type of interaction (H-bond, π-π interaction, charge-charge, and so on) and the amino acid residue involved in the interaction/s.
Response: Table 3 was added in the revised manuscript to conclude the important interactions between the compounds and CSF1R and EGFR to respond this issue.
===================================================
6.In paragraph 2.6is reported the study by MD of potential aggregation in the racemic mixture of (±)-kusunokinin. It is not clear if the authors have performed this study in vacuo or in the presence of explicit solvent molecules. From Figure 11 (B-G) it seems that only one molecule of each kusunokinin enantiomer was used in the calculation. The possibility of aggregation could be between two molecules of the same enantiomer present in the racemic mixture, so the calculation on only one molecule of each enantiomer looks not realistic, and it needs to be performed on at least two molecules of each enantiomer or better on a more large numbers of molecules. In paragraph 4.3 a subparagraph, containing calculation details on the aggregation study, needs to be added.
Response: Thanks for your suggestion and that is very constructive comment. In our study, we performed the MD simulation containing one pair of respective kusunokinin enantiomers under the explicit 0.15 M NaCl solution. The simulation contained one pair of kusunokinin in the TIP3P water molecules. We do agree that the aggregation could be formed via at least two of each enantiomer as the reviewer suggested.
We agreed with the other reviewer and we realized that the simulation looked not realistic and could mislead the reader in case that the reader would cite this article in the future. We thus decided to remove this part from the manuscript. In our opinion, the aggregation is of minor point in the manuscript so without this part, the manuscript still provided the core information as the authors would like to.
===================================================
7.In paragraph 4.1.3the authors used a very large box centered on the entire target molecule of the protein, instead of centering a smaller box on the known binding site for each target protein, given the presence of the ligand in the X-ray structure. Note that in Autodock the dimensions of box are reported in “number of points” and the “grid spacing” should be set (normally by default it is 0.375 Å, which is a quarter of C-C bond length, whereas in Autodock Vina the grid spacing should be set at 1Å showing the size directly in Å). The values 126x126x126 reported in Å (in the experimental part) are not realistic because, the box appears much larger than the target protein. W. P. Feinstein and M. Brylinski report in the article titled “Calculating an optimal box size for ligand docking and virtual screening against experimental and predicted binding pockets” (Journal of Cheminformatics (2015) 7:18. DOI 10.1186/s13321-015-0067-5) a script for calculating an optimal box size. The use of such a large box, in addition to providing less accurate docking results, can insert the ligands at different sites than those that bind the reference ligand present in the X-ray structure. Usually, it is used a box which covers all the protein in the blind docking when the binding site of the ligand is unknown. Why the authors have chosen to use a very large box?
Answer: Thanks for the suggestion. The first of all we the grid point of 120x120x120 in the study not the 126x126x126. It is our fault of typo. We do agree that the large box could diminish the accuracy of the docking process. The reason we chose the 120-120-120 grid points because the grid box covered the whole protein system as
- We do not know which sites of kusunokinin could bind into the interested protein.
- We do not know if the kusunokinin always bind to the protein in the same site of the inhibitor.
As in this case, we used a box that covered the entire protein because, as you mentioned, we assume it is blind docking when the ligand's binding site is unknown, and we did try to vary the different grid points to many proteins. Finally, we decided to use 120x120x120 as this number to ensure that the grid point covered all of the proteins of interest and that the randomization could be performed without biasing the ligand's site. We also added the sentence “The x-y-z grid point numbers of 120-120-120 with the center at the macromolecule were exploited in the docking study. Since we performed binding docking assumption, these chosen x-y-z grid point numbers had covered the protein receptor, ensuring that randomization was done without bias. The grid spacing was set as 0.375 Å. “.
===================================================

Round 2
Reviewer 1 Report
The manuscript was corrected according to the suggestions.